# A Validation of Six Wearable Devices for Estimating Sleep, Heart Rate and Heart Rate Variability in Healthy Adults

**DOI:** 10.3390/s22166317

**Published:** 2022-08-22

**Authors:** Dean J. Miller, Charli Sargent, Gregory D. Roach

**Affiliations:** The Appleton Institute for Behavioural Science, Central Queensland University, Wayville, SA 5034, Australia

**Keywords:** wearable sleep monitor, consumer sleep technology, sleep quality, sleep quantity, sleep staging, autonomic nervous system, autonomic modulation, cardiovascular health, photoplethysmography

## Abstract

The primary aim of this study was to examine the validity of six commonly used wearable devices, i.e., Apple Watch S6, Garmin Forerunner 245 Music, Polar Vantage V, Oura Ring Generation 2, WHOOP 3.0 and Somfit, for assessing sleep. The secondary aim was to examine the validity of the six devices for assessing heart rate and heart rate variability during, or just prior to, night-time sleep. Fifty-three adults (26 F, 27 M, aged 25.4 ± 5.9 years) spent a single night in a sleep laboratory with 9 h in bed (23:00–08:00 h). Participants were fitted with all six wearable devices—and with polysomnography and electrocardiography for gold-standard assessment of sleep and heart rate, respectively. Compared with polysomnography, agreement (and Cohen’s kappa) for two-state categorisation of sleep periods (as sleep or wake) was 88% (κ = 0.30) for Apple Watch; 89% (κ = 0.35) for Garmin; 87% (κ = 0.44) for Polar; 89% (κ = 0.51) for Oura; 86% (κ = 0.44) for WHOOP and 87% (κ = 0.48) for Somfit. Compared with polysomnography, agreement (and Cohen’s kappa) for multi-state categorisation of sleep periods (as a specific sleep stage or wake) was 53% (κ = 0.20) for Apple Watch; 50% (κ = 0.25) for Garmin; 51% (κ = 0.28) for Polar; 61% (κ = 0.43) for Oura; 60% (κ = 0.44) for WHOOP and 65% (κ = 0.52) for Somfit. Analyses regarding the two-state categorisation of sleep indicate that all six devices are valid for the field-based assessment of the timing and duration of sleep. However, analyses regarding the multi-state categorisation of sleep indicate that all six devices require improvement for the assessment of specific sleep stages. As the use of wearable devices that are valid for the assessment of sleep increases in the general community, so too does the potential to answer research questions that were previously impractical or impossible to address—in some way, we could consider that the whole world is becoming a sleep laboratory.

## 1. Introduction

Wearable technologies, or “wearables”, are devices that are worn on the body and provide physiological measures directly to smart devices (e.g., smartphones or tablets). The popularity of wearables is underlined by a market valuation of USD 40 billion in 2020 [1]. Several wearable (and nearable) technologies monitor sleep, heart rate and associated metrics [2,3,4]. Typically, wearables provide these measures based on an alternative to more labour-intensive gold-standard measurements. The gold-standard measurement for sleep is laboratory-based polysomnography (PSG) [5]. PSG acquires signals of brain activity, eye movement and muscle tone to classify sleep stages [6]. However, PSG may not be ideal for monitoring sleep in field settings as it is expensive, labour intensive and requires technical expertise [5]. In such cases, the most commonly utilised alternative to PSG has been actigraphy, i.e., the combined use of activity monitors and sleep/wake diaries [7]. Activity monitors are devices worn on the wrist, such as a watch that contains an accelerometer capable of detecting movement. The devices operate on the principle that movement is correlated with wake and that long periods of inactivity are correlated with sleep [8]. Using this principle, actigraphs provide a binary classification of sleep or wake in 30-s or 1-min intervals (i.e., epochs) [8,9]. Despite actigraphy providing a practical alternative to PSG, some devices are limited to binary sleep/wake detection and require a manual download of the data after the device has been worn for a period of time [7].

Recent advances in wearable technology enable devices to automatically detect sleep in real-time and provide sleep metrics the following morning via a digital platform (e.g., smartphone application) [10]. Wearable companies have also begun providing daily metrics that have typically been exclusive to elite sporting environments (e.g., resting heart rate, heart rate variability) [11]. The passive collection of these measures is less intrusive compared to laboratory-based assessments and is therefore of significant interest to athletes and support staff [12]. However, a major limitation of some wearables is that the underlying technology on which their measures are based has not been scrutinised through peer review [3,13,14,15]. Therefore, the aim of this laboratory-based study was to examine the validity of six wearable devices for assessing sleep, heart rate and heart rate variability. The devices included in this study were chosen because, at least anecdotally, they are produced by six of the most successful/popular wearables companies. To examine validity, measures derived from the wearable devices were compared with gold-standard measures derived from polysomnography and electrocardiography. In the following text, we describe the materials and methods, results, discussion and conclusions associated with this laboratory-based validation study.

## 2. Materials and Methods

### 2.1. Participants

Fifty-three healthy young adults participated in the study (26 female, 27 male, mean ± SD age = 25.4 ± 5.9 years). Participants can be considered as an active population, reporting an average frequency of 7.5 ± 0.4 sessions of exercise per week (399 min; 23% light intensity, 47% moderate intensity, 30% vigorous intensity). Participants were excluded if they reported any existing medical conditions or sleep disorders or had a recent history of shift work and/or transmeridian travel. Participants provided written informed consent and were given a nominal honorarium for their involvement. This study was approved by the CQUniversity Human Research Ethics Committee following the guidelines of the National Health and Medical Research Council (Australia).

### 2.2. Procedure

The study was conducted at The Sleep Lab at CQUniversity’s Appleton Institute for Behavioural Sciences in Wayville, South Australia, from November 2020 to April 2021. The Sleep Lab is comprised of two accommodation suites fitted out as serviced apartments. In total, the suites contain six bedrooms, six bathrooms, two kitchens, two dining rooms, a gymnasium and laundry facilities. The accommodation suites are sound-attenuated, windowless and temperature-controlled, such that during time in bed, background noise was 30 dB, bedrooms were completely dark, and the target temperature was 21–23 °C.

Each participant attended The Sleep Lab on a single night in groups of 4. Participants arrived after dinner in the evening (~20:00 h), were fitted with sleep and heart rate monitoring equipment from 21:30 to 22:30 h, had a 9-h sleep opportunity in bed with lights out from 23:00 to 08:00 h, and departed at ~08:30 h after having the monitoring equipment removed. Each participant was fitted with polysomnography for the gold-standard assessment of sleep, an electrocardiogram for the gold-standard assessment of heart rate and six wearable devices—Apple Watch S6, Garmin Forerunner 245, Polar Vantage V, Oura Ring Generation 2, WHOOP 3.0 and Somfit—for alternative assessment of sleep and heart rate.

### 2.3. Gold-Standard Assessment of Sleep and Heart Rate Metrics

#### 2.3.1. Polysomnography (PSG)

Polysomnography (PSG) was used for the gold-standard assessment of sleep. A standard montage of Grass gold-cup electrodes (AstroMed, Inc., West Warwick, RI, USA) was attached to participants’ scalp, face and body. The montage included three channels of electroencephalography (C4–M1, F4–M1, O2–M1) to assess brain activity, two electrooculograms (left/right outer canthus) to assess eye movements and two submental electromyograms to assess muscle tone. PSG data were recorded directly to data acquisition, storage and analysis systems (Grael, Compumedics, Melbourne, Australia). For each 9-h period of time in bed, PSG records were manually scored in 1080 × 30-s epochs by a trained, experienced sleep technician, using standard criteria [16]. To avoid potential inter-rater differences, all PSG records were scored by a single technician. Each 30-s epoch of time in bed was scored as wake, stage 1, 2 or 3 non-REM sleep (N1, N2, N3) or rapid eye movement sleep (REM), i.e., using a 5-state categorisation system. The scored PSG records were subsequently used to determine the amount of time spent in any stage of sleep (total sleep time) and the amount of wake, N1, N2, N3 and REM for each 9-h period of time in bed. N1 and N2 are lighter stages of sleep and N3—sometimes referred to as slow-wave sleep—is a deeper stage of sleep. REM sleep is sometimes referred to as dreaming sleep, but dreaming also occurs in non-REM sleep, albeit about half as often compared to REM sleep [17].

#### 2.3.2. Electrocardiogram (ECG)

An electrocardiogram (ECG) was used for the gold-standard assessment of heart rate. Participants were fitted with a II-lead ECG (Grael, Compumedics, Melbourne, Australia), with a left-positive electrode positioned on the left side of the torso parallel to the left hip and leg, between either the fifth, sixth, or seventh intercostal spaces on the lower left side of the rib cage, and a right-negative electrode placed three centimetres below the right clavicle, positioned on the torso parallel to the right leg. For each sleep period, R-R intervals from the PSG records were identified and subsequently used to calculate measures of heart rate, i.e., beats per minute, and heart rate variability, i.e., root mean square of the standard deviation of successive heartbeats (ms). The sampling period used for these ECG-determined measures was matched to the sampling period used by each wearable device. HR and HRV analyses were conducted using HRVAnalysis software (version 1, ANSLab Tools, Saint-Étienne, France) [18].

### 2.4. Alternative Assessment of Sleep and Heart Rate Metrics

Six wearable devices were used for alternative assessment of sleep and heart rate. Four of the devices are worn on the wrist similar to a watch, i.e., Apple Watch, Garmin, Polar and WHOOP, Oura is worn on the finger as a ring, and Somfit is worn on the forehead as an adhesive patch. Five of the devices, i.e., Apple Watch, Garmin, Polar, Oura and WHOOP, contain an accelerometer to obtain actigraphy data (physical activity) and green and/or infrared LEDs paired with photodiodes to obtain photoplethysmography data (blood volume). In contrast, Somfit obtains electrophysiological signals related to brain activity, eye movement, muscle tone from electrodes, and blood volume from an LED paired with a photodiode, all contained in a forehead patch. Each of the wearable devices applies their own separate set of proprietary algorithms to these data to assess various sleep metrics, heart rate and heart rate variability.

The system used for the multi-state categorisation of sleep/wake states within a sleep period differed between the wearables such that they categorised epochs within a sleep period into three, four or five states. For 3-state categorisation—used by Apple Watch—epochs were scored as wake (equivalent to PSG wake), light sleep (equivalent to PSG N1 or N2) or deep sleep (equivalent to PSG N3 or REM). For 4-state categorisation—used by Garmin, Polar, Oura and WHOOP—epochs were scored as wake (equivalent to PSG wake), light sleep (equivalent to PSG N1 or N2), deep sleep (equivalent to PSG N3) or REM sleep (equivalent to PSG REM). For 5-state categorisation—used by Somfit—epochs were scored as with PSG, i.e., wake, N1, N2, N3 or REM.

For all wearable devices other than Somfit, the start and end of each sleep period were self-determined by their own proprietary automatic sleep detection algorithms. For all such wearable devices, the self-determined sleep period was within the imposed time in bed for all participants. Any time in bed before each device’s self-determined sleep onset or after each device’s self-determined sleep offset was scored by the researchers as wake. In contrast, Somfit did not have an automatic sleep detection algorithm, so the start and end of each sleep period were manually entered into the Somfit records as 23:00 and 08:00 h, respectively.

All manufacturers of the six wearable devices were invited to provide data directly to the researchers rather than the researchers having to obtain the data via the associated apps. The advantage of data direct from a manufacturer is that it may have a greater level of precision than data available in an app. For example, sleep data are typically scored in 30-s epochs, but some apps output/display the data in 5-min epochs. WHOOP and Somfit accepted the invitation to provide data to the researchers, but the others did not. WHOOP and Somfit staff were naïve to the gold-standard data.

#### 2.4.1. Apple Watch S6 (Apple, Cupertino, CA, USA)

Apple Watch is a wrist-worn smartwatch that assesses various sleep metrics, heart rate and heart rate variability via Sleep Watch—a third-party iOS mobile application. In this study, Sleep Watch was used to score all time within a sleep period as wake, light sleep or deep sleep. Data were extracted with Sleep Watch using the maximum available precision, i.e., in 5-min epochs. The 5-min Sleep Watch epochs were split into 10 × 30-s epochs of the same sleep/wake state and used (a) for comparison with 30-s PSG epochs and (b) to calculate the amount of time spent in any stage of sleep (total sleep time) and the amount of wake, light sleep and deep sleep for each 9-h period of time in bed. Sleep Watch was also used to extract measures of heart rate and heart rate variability for each sleep period (although the exact period of sampling used by Sleep Watch was not specified).

#### 2.4.2. Garmin Forerunner 245 (Garmin, Kansas City, MO, USA)

Garmin is a wrist-worn sports watch that assesses various sleep metrics, heart rate and heart rate variability via the manufacturer’s dedicated mobile applications (iOS and Android). In this study, the iOS version of the Garmin app was used to score all time within a sleep period as wake, light sleep, deep sleep or REM sleep. Data were extracted with the Garmin app using the maximum available precision, i.e., in 1-min epochs. The 1-min Garmin epochs were split into 2 × 30-s epochs of the same sleep/wake state and used (a) for comparison with 30-s PSG epochs and (b) to calculate the amount of time spent in any stage of sleep (total sleep time) and the amount of wake, light sleep, deep sleep and REM sleep for each 9-h period of time in bed. The Garmin app was also used to extract measures of heart rate and heart rate variability for a 3-min period as close as practicable to the start of each sleep period, i.e., ~30–60 min prior to lights out.

#### 2.4.3. Polar Vantage V (Polar, Kempele, Finland)

Polar is a wrist-worn sports watch that assesses various sleep metrics, heart rate and heart rate variability via the manufacturer’s dedicated mobile applications (iOS and Android). In this study, the iOS version of the Polar app was used to score all time within a sleep period as wake, light sleep, deep sleep or REM sleep. Data were extracted with the Polar app using the maximum available precision, i.e., in 30-s epochs, and used (a) for comparison with 30-s PSG epochs and (b) to calculate the amount of time spent in any stage of sleep (total sleep time) and the amount of wake, light sleep, deep sleep and REM sleep for each 9-h period of time in bed. The Polar app was used to extract measures of heart rate and heart rate variability for a 4-h period within each sleep period.

#### 2.4.4. Oura Ring Generation 2 (Oura, Oulu, Finland)

Oura is a finger-worn smart ring that assesses various sleep metrics, heart rate and heart rate variability via the manufacturer’s dedicated mobile applications (iOS and Android). In this study, the iOS version of the Oura app was used to score all time within a sleep period as wake, light sleep, deep sleep or REM sleep. The 5-min Oura epochs were split into 10 × 30-s epochs of the same sleep/wake state and used (a) for comparison with 30-s PSG epochs and (b) to calculate the amount of time spent in any stage of sleep (total sleep time) and the amount of wake, light sleep, deep sleep and REM sleep for each 9-h period of time in bed. The Oura app was also used to extract measures of heart rate and heart rate variability for each sleep period (although the exact period of sampling used by the Oura app was not specified).

#### 2.4.5. WHOOP 3.0 (WHOOP, Boston, MA, USA)

WHOOP is a wrist-worn fitness tracker that assesses various sleep metrics, heart rate and heart rate variability via the manufacturer’s dedicated mobile applications (iOS and Android). Typically, the WHOOP app scores all time within a sleep period as wake, light sleep, deep sleep or REM sleep. For this study, these data were provided to the researchers by staff at WHOOP who were naïve to the gold-standard PSG data. Data were provided by WHOOP using the maximum available precision, i.e., in 30-s epochs, and used (a) for comparison with 30-s PSG epochs, and (b) to calculate the amount of time spent in any stage of sleep (total sleep time) and the amount of wake, light sleep, deep sleep and REM sleep for each 9-h period of time in bed. For this study, R-R intervals for the whole duration of each sleep period were provided to the researchers by staff at WHOOP who were naïve to the gold-standard ECG data. The researchers used these data to calculate average heart rate and average heart rate variability for each sleep period.

#### 2.4.6. Somfit (Compumedics, Melbourne, Australia)

Somfit is a device that attaches to the forehead via an adhesive patch and can be used to assess various sleep metrics, heart rate and heart rate variability via the manufacturer’s dedicated mobile applications (iOS and Android). Typically, the Somfit app scores all time within a period of time in bed as wake, N1, N2, N3 or REM. For this study, these data were provided to the researchers by staff at Somfit who were naïve to the gold-standard PSG data. Data were provided by Somfit using the maximum available precision, i.e., in 30-s epochs, and used (a) for comparison with 30-s PSG epochs and (b) to calculate the amount of time spent in any stage of sleep (total sleep time) and the amount of wake, N1, N2, N3 and REM for each 9-h period of time in bed. In addition, heart rate and heart rate variability data, sampled at 125 Hz for the whole duration of each sleep period, were provided to the researchers by staff at Somfit who were naïve to the gold-standard ECG data. The researchers used these data to calculate average heart rate and average heart rate variability for each sleep period.

### 2.5. Data Analysis

The analysis of data followed a standardised framework for assessing the performance of sleep-related wearable devices [19].

#### 2.5.1. Sleep—Epoch-by-Epoch Comparisons

Epoch-by-epoch comparisons between the wearable devices and the gold-standard PSG were performed for both 2-state categorisation of time in bed (as sleep or wake) and multi-state categorisation of time in bed (as a particular sleep stage or wake).

For the 2-state categorisation of sleep, each 30-s epoch was classified as one of four types based on the agreement (or not) between each wearable device and PSG (Table 1), and the following variables were calculated for each wearable device:Sensitivity for sleep (%) = TS/(TS + FW) × 100, i.e., the percentage of PSG sleep epochs correctly scored as sleep by the wearable device.Sensitivity for wake (%) = TW/(TW + FS) × 100, i.e., the percentage of PSG wake epochs correctly scored as wake by the wearable device (sometimes referred to as specificity).Agreement (%) = (TS + TW)/(TS + TW + FS + FW) × 100, i.e., the percentage of all PSG epochs correctly scored as sleep or wake by the wearable device.

For the most common multi-state categorisation of sleep (i.e., 4-state categorisation), each 30-s epoch was classified as one of sixteen types based on the agreement (or not) between each wearable device and PSG (Table 2), and the following variables were calculated for each wearable device:Sensitivity ^A^ for light sleep (%) = TL/(TL + FW_N1N2_ + FD_N1N2_ + FR_N1N2_) × 100, i.e., the percentage of PSG N1 or N2 epochs correctly scored as light sleep by the wearable device.Sensitivity ^B^ for deep sleep (%) = TD/(TD + FW_D_ + FL_D_ + FR_D_) × 100, i.e., the percentage of PSG N3 epochs correctly scored as deep sleep by the wearable device.Sensitivity ^B^ for REM sleep (%) = TR/(TR + FW_R_ + FL_R_ + FD_R_) × 100, i.e., the percentage of PSG REM epochs correctly scored as REM sleep by the wearable device.Sensitivity for wake (%) = TW/(TW + FL_W_ + FD_W_ + FR_W_) × 100, i.e., the percentage of PSG wake epochs correctly scored as wake by the wearable device (sometimes referred to as specificity).Agreement (%) = (TW + TL + TD + TR)/(TW + TL + TD + TR + FW_N1N2_ + FW_N3_ + FW_R_ + FL_W_ + FL_N3_ + FL_R_ + FD_W_ + FD_N1N2_ + FD_R_ + FR_W_ + FR_N1N2_ + FR_N3_) × 100, i.e., the percentage of all PSG epochs correctly scored as light sleep, deep sleep, REM sleep or wake, by the wearable device.

^A^ Somfit only—Somfit did not combine N1 and N2 into a single state of light sleep, so it was possible to calculate its separate sensitivity for N1 and N2.^B^ Apple Watch only—Apple Watch combined N3 and REM into a single state of deep sleep, so its sensitivity for deep sleep was calculated as the percentage of PSG N3 or REM epochs correctly scored as deep sleep.

Sensitivity and agreement indicate the likelihood that a PSG epoch will be correctly identified by a wearable device. Cohen’s kappa (κ) was also calculated to evaluate the agreement values relative to that which could be expected due to chance [20], and the statistic was interpreted using standard guidelines where: 0–0.20 = slight agreement; 0.21–0.40 = fair agreement; 0.41–0.60 = moderate agreement; 0.61–0.80 = substantial agreement; 0.81–0.99 = almost perfect agreement; and 1 = perfect agreement [19].

Prior to the start of each night of data collection, clock time was manually synchronised on all gold-standard equipment and wearable devices. Due to the vagaries of apps associated with the wearable devices, synchronisation may be imperfect such that there may be minor differences in clock time between some pairs of gold-standard/wearable sleep records. To minimise the effect of these differences, a 2-state agreement was examined with offset adjustments of 5 × 30-s epochs in both directions for each wearable device at the individual level, and the offset with the highest agreement value was applied.

#### 2.5.2. Sleep—Bland-Altman Analyses, Bias and Absolute Bias

Agreement between the wearable devices and PSG for total sleep time was examined using the limits of agreement method for repeated measurements [21]. For each wearable device, modified Bland–Altman plots were produced to display (a) the pairwise differences between the wearable- and PSG-derived values for total sleep time, (b) the mean difference between the wearable- and PSG-derived values for total sleep time (bias), and (c) the 95% limits of agreement, i.e., bias ± (1.96 × SD). Plots were examined for heteroscedasticity and proportional bias using the Breusch–Pagan test and ordinary least squares regression, respectively. In cases where heteroscedasticity and/or proportional bias were present, the bias and 95% limits of agreement were adjusted accordingly. For each wearable device, the mean difference (bias) and mean absolute difference (absolute bias) between the wearable- and PSG-derived values for total sleep time, time awake and time in each sleep state were also calculated. For total sleep time, wearable-based estimates are typically considered to be clinically satisfactory if the absolute bias is <30 min [7].

#### 2.5.3. Heart Rate and Heart Rate Variability—Bland–Altman Analyses, Mean Bias and Absolute Bias

Agreement between the wearable devices and ECG for heart rate and heart rate variability were examined as for total sleep time above, i.e., using Bland–Altman plots with pairwise differences, biases and 95% limits of agreement, and using the Breusch–Pagan test for heteroscedasticity and proportional bias [22]. Intraclass correlation coefficients were also calculated and interpreted using standard guidelines where: <0.40 = poor; 0.40–0.59 = fair; 0.60–0.74 = good; and 0.75–1.00 = excellent [23].

## 3. Results

### 3.1. Apple Watch S6

Compared with PSG for the two-state categorisation of sleep/wake, Apple Watch correctly identified 97% of PSG sleep epochs, 26% of PSG wake epochs and 88% of all PSG epochs, with a kappa value of 0.30, which indicates a fair level of agreement (Table 3). Apple Watch overestimated total sleep time by an average of 39.5 min (Table 4), its absolute bias was 48.1 min (Table 4), and it had proportional bias and heteroscedasticity (Figure 1). Proportional bias manifested as the degree of overestimation in total sleep time being greater for poorer-quality sleep, i.e., those with less time in bed converted to sleep, than for better-quality sleep, i.e., those with greater time in bed converted to sleep—with bias ranging from 104.7 to 10.1 min. Heteroscedasticity manifested as the limits of agreement around the bias being wider for poorer-quality sleep (−19.0 to 227.7 min) than for better-quality sleep (−18.2 to 39.2 min). Compared with PSG for multi-state categorisation of sleep/wake, Apple Watch correctly identified 44% of PSG light sleep epochs, 71% of PSG epochs scored as stage 3 non-REM sleep or REM sleep, 26% of PSG wake epochs and 53% of all PSG epochs, with a kappa value of 0.20, which indicates a slight level of agreement (Table 3). The main sources of error were that Apple Watch misclassified 51% of PSG wake as light sleep, 52% of PSG N1 or N2 as deep sleep and 26% of PSG N3 or REM as light sleep (Table 5).

Compared with ECG-derived heart rate (HR), Apple Watch overestimated HR by an average of 0.5 beats per min, absolute bias was 1.5 beats per min, intraclass correlation was 0.96, i.e., excellent (Table 6), and it had non-proportional bias and homoscedasticity, with limits of agreement around the bias at −3.5 and 4.6 beats per min (Figure 2).

Compared with ECG-derived heart rate variability (HRV), Apple Watch underestimated HRV by an average of 9.6 ms, absolute bias was 22.5 ms, and intraclass correlation was 0.67, i.e., good (Table 6). Apple Watch estimates had proportional bias with heteroscedasticity such that HRV was overestimated at lower ECG-derived values and underestimated at higher ECG-derived values, with bias ranging from 14.6 to −47.6 ms, and limits of agreement were narrower at lowest ECG-derived values (−10.8 to 40.0 ms) and wider at highest ECG-derived values (−123.6 to 28.4 ms) (Figure 3).

### 3.2. Garmin Forerunner 245 Music

Compared with PSG for two-state categorisation of sleep/wake, Garmin correctly identified 98% of sleep epochs, 27% of wake epochs and 89% of all epochs, with a kappa value of 0.35, which indicates a fair level of agreement (Table 3). Garmin overestimated total sleep time by an average of 43.8 min (Table 4), its absolute bias was 45.3 min (Table 4), and it had proportional bias and heteroscedasticity (Figure 1). Proportional bias manifested as the degree of overestimation in total sleep time being greater for poorer-quality sleep, i.e., those with less time in bed converted to sleep, than for better-quality sleep, i.e., those with greater time in bed converted to sleep-with bias ranging from 125.0 to 10.1 min. Heteroscedasticity manifested as the limits of agreement around the bias being wider for poorer-quality sleep (−49.1 to 299.0 min) than for better-quality sleep (3.6 to 16.7 min). Compared with PSG for multi-state categorisation of sleep/wake, Garmin correctly identified 68% of PSG light sleep epochs, 28% of PSG deep sleep epochs, 50% of PSG REM sleep epochs, 27% of PSG wake epochs and 50% of all PSG epochs, with a kappa value of 0.25, which indicates a fair level of agreement (Table 3). The main sources of error were that Garmin misclassified 45% of PSG wake as light sleep, 21% of PSG N1 or N2 as REM, 61% of PSG N3 as light sleep and 42% of PSG REM as light sleep (Table 5).

Compared with ECG-derived heart rate (HR), Garmin overestimated HR by an average of 5.0 beats per min, absolute bias was 5.4 beats per min, intraclass correlation was 0.41, i.e., fair (Table 6), and it had non-proportional bias and homoscedasticity, with limits of agreement around the bias at –20.0 and 30.0 beats per min (Figure 2).

Compared with ECG-derived heart rate variability (HRV), Garmin underestimated HRV by an average of 22.4 ms, absolute bias was 33.1 ms, and intraclass correlation was 0.24, i.e., poor (Table 6). Garmin estimates had proportional bias with homoscedasticity such that HRV was overestimated at lower ECG-derived values and underestimated at higher ECG-derived values, with bias ranging from 15.2 to −146.3 ms, and limits of agreement were ±44.7 ms across the range of ECG-derived values (Figure 3).

### 3.3. Polar Vantage V

Compared with PSG for two-state categorisation of sleep/wake, Polar correctly identified 92% of sleep epochs, 51% of wake epochs and 87% of all epochs, with a kappa value of 0.44, which indicates a moderate level of agreement (Table 3). Polar underestimated total sleep time by an average of 0.8 min (Table 4), its absolute bias was 31.2 min (Table 4), and it had non-proportional bias with heteroscedasticity (Figure 1). Heteroscedasticity manifested as the limits of agreement around the constant bias for total sleep time being wider for poorer-quality sleep, i.e., those with less time in bed converted to sleep (−197.7 to 196.1 min), than for better-quality sleep, i.e., those with more time in bed converted to sleep (−17.2 to 16.6 min). Compared with PSG for multi-state categorisation of sleep/wake, Polar correctly identified 60% of PSG light sleep epochs, 33% of PSG deep sleep epochs, 49% of PSG REM sleep epochs, 51% of PSG wake epochs and 51% of all PSG epochs, with a kappa value of 0.28, which indicates a fair level of agreement (Table 3). The main sources of error were that Polar misclassified 31% of PSG wake as light sleep, 17% of PSG N1 or N2 as REM, 53% of PSG N3 as light sleep and 43% of PSG REM as light sleep (Table 5).

Compared with ECG-derived heart rate (HR), Polar underestimated HR by an average of 1.1 beats per min, absolute bias was 1.5 beats per min, intraclass correlation was 0.93, i.e., excellent (Table 6), and it had non-proportional bias and homoscedasticity, with limits of agreement around the bias at −5.5 and 3.3 beats per min (Figure 2).

Compared with ECG-derived heart rate variability (HRV), Polar underestimated HRV by an average of 8.7 ms, absolute bias was 18.8 ms, and intraclass correlation was 0.65, i.e., good (Table 6). Polar estimates had proportional bias with heteroscedasticity, such that HRV was overestimated at lower ECG-derived values and underestimated at higher ECG-derived values, with bias ranging from 28.8 to −124.2 ms, and limits of agreement were narrower at lowest ECG-derived values (11.6 to 45.9 ms) and wider at highest ECG-derived values (−223.8 to −24.6 ms) (Figure 3).

### 3.4. Oura Ring Generation 2

Compared with PSG for two-state categorisation of sleep/wake, Oura correctly identified 94% of sleep epochs, 57% of wake epochs and 89% of all epochs, with a kappa value of 0.51, which indicates a moderate level of agreement (Table 3). Oura overestimated total sleep time by an average of 1.5 min (Table 4), its absolute bias was 29.0 min (Table 4), and it had proportional bias with homoscedasticity (Figure 1). Proportional bias manifested as an overestimation of total sleep time for poorer-quality sleep, i.e., those with less time in bed converted to sleep, and an underestimation of total sleep time for better-quality sleep, i.e., those with greater time in bed converted to sleep—with bias ranging from 49.8 to −20.0 min. Limits of agreement around the bias were constant across all values of total sleep time at ±73.6 min. Compared with PSG for multi-state categorisation of sleep/wake, Oura correctly identified 66% of PSG light sleep epochs, 62% of PSG deep sleep epochs, 52% of PSG REM sleep epochs, 57% of PSG wake epochs and 61% of all PSG epochs, with a kappa value of 0.43, which indicates a moderate level of agreement (Table 3). The main sources of error were that Oura misclassified 30% of PSG wake as light sleep, 15% of PSG N1 or N2 as deep sleep, 32% of PSG N3 as light sleep and 37% of PSG REM as light sleep (Table 5).

Compared with ECG-derived heart rate (HR), Oura overestimated HR by an average of 0.1 beats per min, absolute bias was 1.80 beats per min, intraclass correlation was 0.85, i.e., excellent (Table 6), and it had non-proportional bias and homoscedasticity, with limits of agreement around the bias at −8.7 and 8.8 beats per min (Figure 2).

Compared with ECG-derived heart rate variability (HRV), Oura underestimated HRV by an average of 10.2 ms, absolute bias was 18.9 ms, and intraclass correlation was 0.63, i.e., good (Table 6). Oura estimates had proportional bias with heteroscedasticity such that HRV was overestimated at lower ECG-derived values and underestimated at higher ECG-derived values, with bias ranging from 25.9 to −111.0 ms, and limits of agreement were narrower at lowest ECG-derived values (9.8 to 42.1 ms) and wider at highest ECG-derived values (−248.6 to 26.7 ms) (Figure 3).

### 3.5. WHOOP 3.0

Compared with PSG for two-state categorisation of sleep/wake, WHOOP correctly identified 90% of sleep epochs, 56% of wake epochs and 86% of all epochs, with a kappa value of 0.44, which indicates a moderate level of agreement (Table 3). WHOOP underestimated total sleep time by an average of 12.2 min (Table 4), its absolute bias was 30.3 min (Table 4), and it had proportional bias with homoscedasticity (Figure 1). Proportional bias manifested as an overestimation of total sleep time for poorer-quality sleep, i.e., those with less time in bed converted to sleep, and an underestimation of total sleep time for better-quality sleep, i.e., those with greater time in bed converted to sleep—with bias ranging from 52.2 to −34.4 min. Limits of agreement around the bias were constant across all values of total sleep time at ±61.2 min. Compared with PSG for multi-state categorisation of sleep/wake, WHOOP correctly identified 58% of PSG light sleep epochs, 62% of PSG deep sleep epochs, 66% of PSG REM sleep epochs, 56% of PSG wake epochs and 60% of all PSG epochs, with a kappa value of 0.44, which indicates a moderate level of agreement (Table 3). The main sources of error were that WHOOP misclassified 28% of PSG wake as light sleep, 20% of PSG N1 or N2 as REM, 32% of PSG N3 as light sleep and 23% of PSG REM epochs as light sleep (Table 5).

Compared with ECG-derived heart rate (HR), WHOOP underestimated HR by an average of 0.3 beats per min, absolute bias was 0.7 beats per min, intraclass correlation was 0.99, i.e., excellent (Table 6), and it had non-proportional bias and homoscedasticity, with limits of agreement around the bias at −2.3 and 1.6 beats per min (Figure 2).

Compared with ECG-derived heart rate variability (HRV), WHOOP underestimated HRV by an average of 4.5 ms, absolute bias was 4.7 ms, intraclass correlation was 0.99, i.e., excellent (Table 6), and it had non-proportional bias with homoscedasticity, with limits of agreement around the bias at −2.9 and 12.3 ms (Figure 3).

### 3.6. Somfit

Compared with PSG for two-state categorisation of sleep/wake, Somfit correctly identified 92% of sleep epochs, 57% of wake epochs and 87% of all epochs, with a kappa value of 0.48, which indicates a moderate level of agreement (Table 3). Somfit underestimated total sleep time by an average of 5.5 min (Table 4), its absolute bias was 33.7 min (Table 4), and it had non-proportional bias and homoscedasticity, with limits of agreement around the bias at −93.4 and 82.4 min (Figure 1). Compared with PSG for multi-state categorisation of sleep/wake, Somfit correctly identified 1% of PSG N1 sleep epochs, 79% of PSG N2 sleep epochs, 65% of PSG deep sleep epochs, 58% of PSG REM sleep epochs, 57% of PSG wake epochs and 65% of all PSG epochs, with a kappa value of 0.52, which indicates a moderate level of agreement (Table 3). The main sources of error were that Somfit misclassified 24% of PSG wake as N1, 53% of PSG N1 as N2, 25% of PSG N3 as N2 and 26% of PSG REM as N2 (Table 5).

Compared with ECG-derived heart rate (HR), Somfit overestimated HR by an average of 2.2 beats per min, absolute bias was 2.6 beats per min, and intraclass correlation was 0.65, i.e., good (Table 6). Somfit estimates had proportional bias with homoscedasticity, such that HRV was overestimated at lower ECG-derived values and underestimated at higher ECG-derived values, with bias ranging from 7.5 to −5.9 beats per minute, and limits of agreement were ±11.3 beats per min across the range of ECG-derived values (Figure 2).

Compared with ECG-derived heart rate variability (HRV), Somfit underestimated HRV by an average of 20.5 ms, absolute bias was 24.0 ms, and intraclass correlation was 0.69, i.e., good (Table 6). Somfit estimates had proportional bias with heteroscedasticity such that HRV was overestimated at lower ECG-derived values and underestimated at higher ECG-derived values, with bias ranging from 24.5 to −157.2 ms, and limits of agreement were narrower at lowest ECG-derived values (9.1 to 39.9 ms) and wider at highest ECG-derived values (−235.4 to −78.9 ms) (Figure 3).

## 4. Discussion

### 4.1. Sleep

The majority of the devices included in this study performed similarly to previous validations examining the capability of devices to estimate two-state sleep (i.e., agreement) [10,15,21]. All devices detected >90% of sleep epochs (i.e., sensitivity), but Polar, Oura Gen 2, WHOOP 3.0 and Somfit outperformed Apple Watch and Garmin for detecting wake epochs (i.e., specificity). Low specificity, or high variability in specificity, is a common finding when validating devices that rely predominantly on actigraphy to estimate sleep [9,10,21,24]. The detection of wake within sleep is difficult due to the similarity in movement between restful wake and sleep. Therefore, it is reasonable to suggest that devices that are better at detecting wake within sleep have refined their proprietary algorithms to detect wake using metrics other than movement (e.g., heart rate, heart rate variability).

The devices included in this study ranged from 50 to 65% agreement for multi-state sleep when compared to PSG. Therefore, it is clear that all of these devices can improve their estimation of multi-state sleep. It is important to consider that some of the devices included in this study estimate multi-state sleep differently. For polysomnographic sleep records, sleep is categorised into five states: wake (W), stage 1 (N1), stage 2 (N2), stage 3 (N3), and REM (R). Most of the devices (i.e., Garmin, Polar, Oura Gen 2, WHOOP 3.0) classify sleep into four stages: wake (W), light sleep (N1 and N2), deep sleep (N3), and REM (R). Apple Watch classifies sleep into three stages: wake (W), light sleep (N1 and N2), and deep sleep (N3 and REM). Somfit classified sleep in the same structure as PSG (i.e., W, N1, N2, N3, R). The agreement for multi-state sleep ranged from 50 to 65%, but it is important to contextualise the agreement relative to PSG. PSG scoring is performed manually by trained technicians, and there is often variability across technicians scoring identical records—the level of chance-corrected agreement between technicians is substantial rather than almost perfect (κ = 0.78) [20]. Given this benchmark, the devices with fair or moderate chance-corrected agreement appear to provide reasonable estimates of multi-state sleep but need improvement to reach trained technician levels of agreement. However, it is reasonable to suggest that the better-performing devices for estimating multi-state sleep provide valuable information when monitoring for sustained, meaningful changes in sleep stage. Therefore, the sleep-related practical applications from this study are (1) the included devices can be used as an alternative to PSG for estimation of two-state sleep; and (2) the devices with higher relative agreement for multi-state sleep, i.e., Oura Gen 2, WHOOP 3.0 and Somfit, may be used to monitor for sustained, meaningful changes in sleep architecture (i.e., time spent in different stages of sleep).

### 4.2. Heart Rate and Heart Rate Variability

For measurement of heart rate, the devices ranged from moderate relative agreement to almost perfect relative agreement with ECG. This outcome is to be expected given that the wearable devices examined in the present study utilise a validated method of estimated heart rate (i.e., photoplethysmography). Furthermore, heart rate is a primary metric obtained from photoplethysmography that requires no additional algorithms to estimate. Therefore, the devices with moderate to high relative agreement can be used to monitor heart rate.

The way in which heart rate variability was sampled differed across the devices. Of the devices examined in the present study, Apple Watch, Oura Gen 2, WHOOP 3.0 and Somfit provided measures based on the entire sleep period, Polar provided measures sampled over 4-h periods within sleep, and Garmin provided measures during a 3-min manual test during wake. Devices that sampled heart rate during sleep had high relative agreement for heart rate estimation and moderate to high relative agreement for heart rate variability. It is important to consider that heart rate metrics for four of the devices, i.e., Apple Watch, Garmin, Polar and Oura Gen 2, were obtained using the manufacturers’ proprietary algorithms. Therefore, these comparisons are “naive” and based on the best publicly available information. The lowest agreements for heart rate and heart rate variability were observed during the HRV test conducted with the Garmin device. It is possible that sampling during wake may have contributed to the low agreement with ECG and that Garmin devices capable of sampling during sleep may have better agreement. The one device for which raw R-R intervals were provided, i.e., WHOOP 3.0, had an almost perfect agreement for measures of both heart rate and heart rate variability. The findings of this study highlight varying levels of validity for the measurement of heart rate and heart rate variability by wearable devices. The better-performing devices appear to provide valid measures of heart rate and heart rate variability in the absence of gold-standard ECG.

### 4.3. Boundary Conditions

There are a number of boundary conditions that should be considered when interpreting the results of this study. It is important to understand that the analyses were comparisons of six independent devices to gold-standard PSG and ECG rather than a comparison between devices. Indeed, the results indicate that some devices may have a higher agreement with gold-standard measurements than others, but differences in data acquisition between devices should be considered. For the majority of the devices (i.e., except WHOOP and Somfit), data were exported directly from the manufacturers’ mobile applications. In some cases, data could only be exported in 5-min epochs rather than 30-s epochs. This lack of precision may have an impact on agreement with PSG and ECG.

It is unclear how these devices may perform in clinical populations or in environments where there may be a high proportion of wake within sleep. Future research should examine the validity of wearable devices to estimate sleep, heart rate and heart rate variability in differing populations (e.g., patients with sleep disorders) and under different circumstances (e.g., HRV validation when supine vs. sitting).

## 5. Conclusions

Analyses regarding the two-state categorisation of time in bed (as sleep or wake) indicate that all six wearable devices included in this study are valid for the field-based assessment of total sleep time. In contrast, analyses regarding the multi-state categorisation of sleep (as a specific sleep stage or wake) indicate that all six devices require further improvement for the assessment of specific sleep stages. As the use of wearable devices that are valid for the assessment of sleep increases in the general community, so too does the potential to answer research questions that were previously impractical or impossible to address—in some way, we could consider that the whole world is becoming a sleep laboratory.

By the nature of the market for wearable devices that assess health-related metrics, new versions and variants are being continuously developed by providers and released to consumers. This presents a challenge for those who are working to conduct independent validation studies. However, it seems reasonable to assume that newer models will perform at least as well, if not better, than older models from the same provider when compared with the relevant gold standard(s). Therefore, the data presented here should not be considered obsolete when the models examined here have been usurped by newer models. Rather, these data should be used as the best indication of the likely performance of any newer models—until they themselves have been subject to independent validation studies.

## Figures and Tables

**Figure 1 sensors-22-06317-f001:**
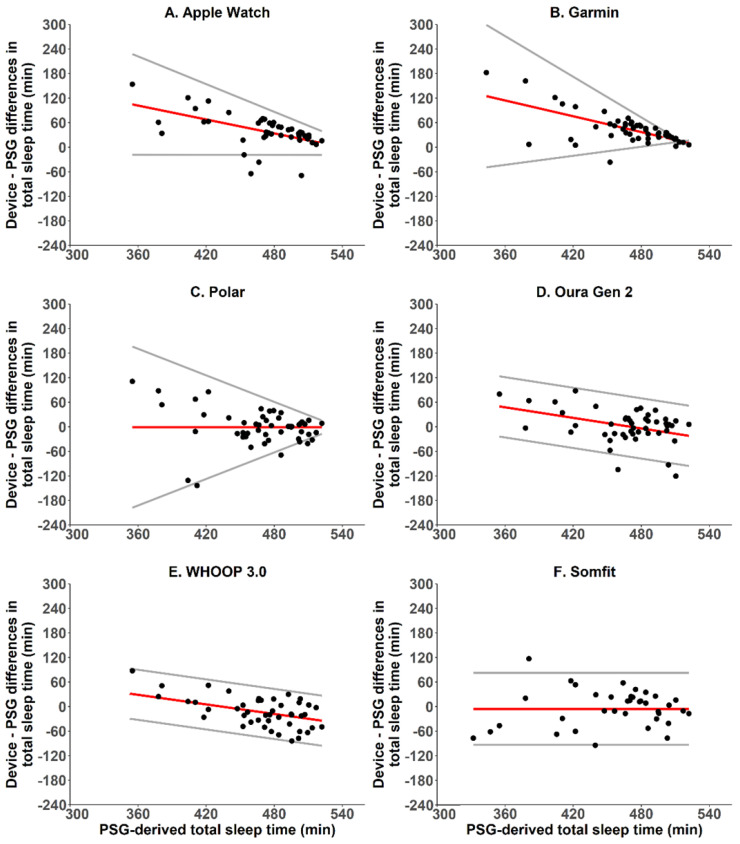
Bland–Altman plots for device–derived and PSG–derived measures of total sleep time. The y-axes represent the difference between the measures—positive values indicate devices overestimate relative to PSG, and negative values indicate devices underestimate relative to PSG. Solid red lines indicate the device bias relative to PSG. Solid grey lines indicate the 95% limits of agreement (±1.96 standard deviation).

**Figure 2 sensors-22-06317-f002:**
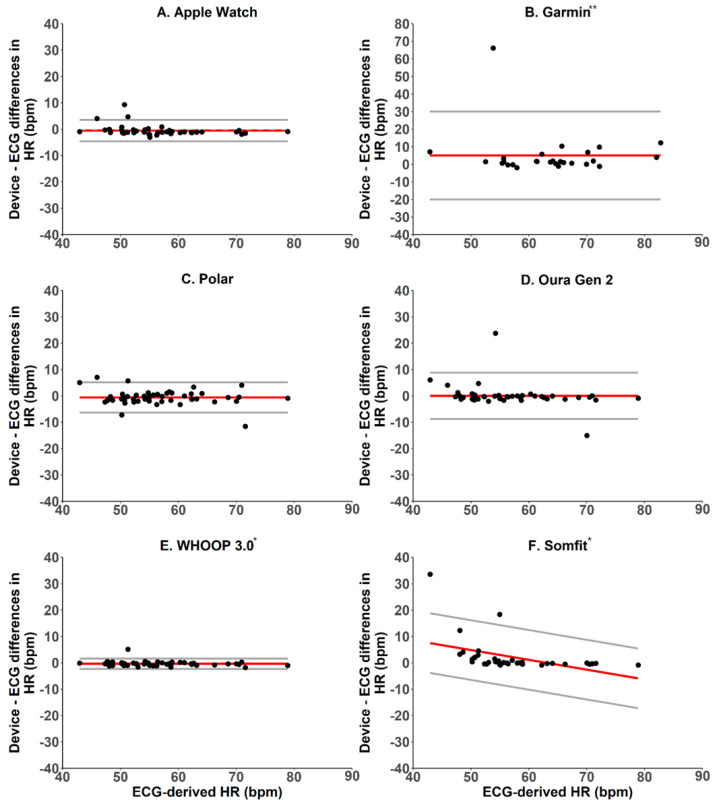
Bland–Altman plots for device–derived and PSG–derived measures of heart rate. The y-axes represent the difference between the measures—positive values indicate devices overestimate relative to PSG, and negative values indicate devices underestimate relative to PSG. Solid red lines indicate the device bias relative to PSG. Solid grey lines indicate the 95% limits of agreement (±1.96 standard deviation). * manufacturers provided raw data; ** y-axis was adjusted in plot B to accommodate for one data point outside of the common y–axis range.

**Figure 3 sensors-22-06317-f003:**
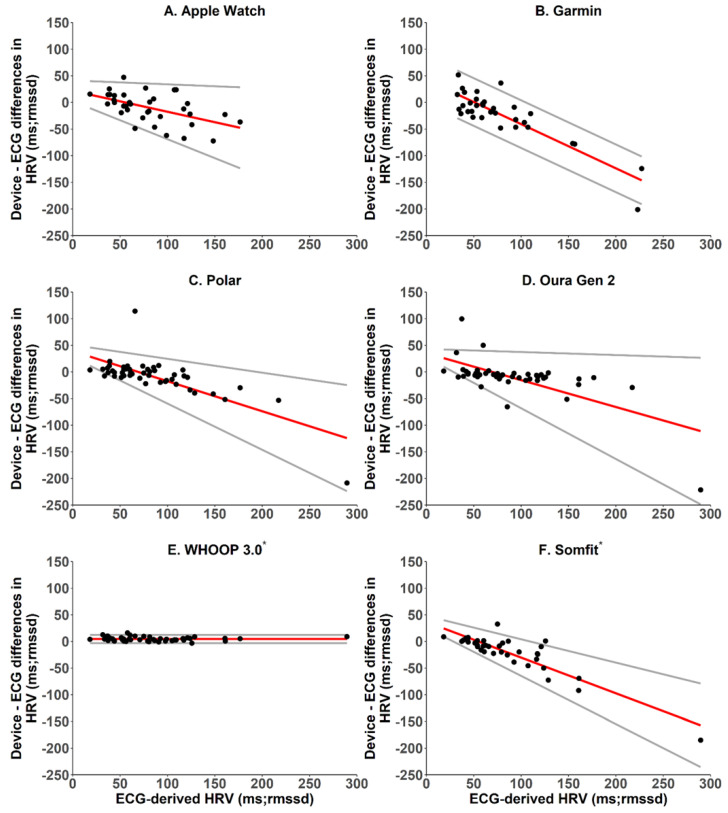
Bland–Altman plots for device–derived and PSG–derived measures of heart rate variability. The y–axes represent the difference between the measures—positive values indicate devices overestimate relative to PSG, and negative values indicate devices underestimate relative to PSG. Solid red lines indicate the device bias relative to PSG. Solid grey lines indicate the 95% limits of agreement (±1.96 standard deviation). * Manufacturers provided raw data.

**Table 1 sensors-22-06317-t001:** Sleep/wake agreement matrix.

		Wearable Device	
		Sleep	Wake
PSG	Sleep	True Sleep (TS)	False Wake (FW)
	Wake	False Sleep (FS)	True Wake (TW)

**Table 2 sensors-22-06317-t002:** Sleep stage agreement matrix.

		Wearable Device (Garmin, Polar, Oura, WHOOP)			
		Wake	Light	Deep	REM
PSG	Wake	True Wake (TW)	False Light (FL_W_)	False Deep (FD_W_)	False REM (FR_W_)
	N1 or N2	False Wake (FW_N1N2_)	True Light (TL)	False Deep (FD_N1N2_)	False REM (FR_N1N2_)
	N3	False Wake (FW_N3_)	False Light (FL_N3_)	True Deep (TD)	False REM (FR_N3_)
	REM	False Wake (FW_R_)	False Light (FL_R_)	False Deep (FD_R_)	True REM (TR)

**Table 3 sensors-22-06317-t003:** Epoch-by-epoch statistics between PSG and each device.

Variable	Apple Watch	Garmin	Polar	Oura (Gen.2)	WHOOP (3.0)	Somfit
Two-state Analysis:						
Sensitivity for sleep (%)	97	98	92	94	90	92
Sensitivity for wake (%)	26	27	51	57	56	57
Agreement (%)	88	89	87	89	86	87
Cohen’s Kappa (k)	0.30	0.35	0.44	0.51	0.44	0.48
Multi-state Analysis:						
Sensitivity for N1 sleep (%)	–	–	–	–	–	1
Sensitivity for N2 sleep (%)	–	–	–	–	–	79
Sensitivity for light sleep (%)	44	68	60	66	58	–
Sensitivity for deep sleep (%)	–	28	33	62	62	65
Sensitivity for REM sleep (%)	–	50	49	52	66	58
Sensitivity for deep/REM sleep (%)	71	–	–	–	–	–
Sensitivity for wake (%)	26	27	51	57	56	57
Agreement (%)	53	50	51	61	60	65
Cohen’s Kappa (k)	0.20	0.25	0.28	0.43	0.44	0.52

Note: An empty cell indicates that the device did not provide that variable.

**Table 4 sensors-22-06317-t004:** Biases between PSG and each device for sleep metrics. Data are mean ± SD.

Variable	Apple Watch	Garmin	Polar	Oura (Gen.2)	WHOOP (3.0)	Somfit
Bias:						
Total sleep time (min)	39.5 ± 41.5	43.8 ± 38.0	−0.8 ± 45.3	1.5 ± 40.9	−12.2 ± 36.3	−5.5 ± 44.9
N1 sleep (min)	–	–	–	–	–	−35.7 ± 17.1
N2 sleep (min)	–	–	–	–	–	57.1 ± 48.4
Light sleep (min)	−35.4 ± 68.1	76.0 ± 57.4	41.8 ± 53.2	19.8 ± 57.4	−15.6 ± 50.7	–
Deep sleep (min)	–	−52.4 ± 64.2	−40.4 ± 41.7	2.4 ± 56.4	−19.6 ± 34.3	−11.7 ± 32.2
REM sleep (min)	–	20.2 ± 64.1	−2.2 ± 34.6	−20.7 ± 35.3	22.9 ± 45.4	−15.2 ± 34.6
Deep/REM sleep (min)	74.9 ± 73.8	–	–	–	–	–
Wake (min)	−38.7 ± 33.2	−44.2 ± 36.2	−21.3 ± 34.6	−3.1 ± 36.1	13.1 ± 32.6	11.2 ± 43.2
Absolute Bias:						
Total sleep time (min)	48.1 ± 30.4	45.3 ± 36.3	31.2 ± 32.6	29.0 ± 28.6	30.3 ± 23.0	33.7 ± 27.1
N1 sleep (min)	–	–	–	–	–	35.7 ± 17.1
N2 sleep (min)	–	–	–	–	–	61.6 ± 42.3
Light sleep (min)	59.4 ± 48.1	81.6 ± 48.9	52.7 ± 42.1	48.8 ± 35.5	43.6 ± 29.5	–
Deep sleep (min)	–	72.4 ± 39.8	47.7 ± 32.8	42.4 ± 36.8	30.6 ± 24.4	28.5 ± 18.9
REM sleep (min)	–	54.5 ± 38.6	26.1 ± 22.5	32.8 ± 24.3	41.9 ± 28.5	29.5 ± 23.2
Deep/REM sleep (min)	86.8 ± 58.9	–	–	–	–	–
Wake (min)	42.0 ± 28.9	45.7 ± 34.3	26.0 ± 31.1	25.0 ± 26.0	28.0 ± 20.9	34.0 ± 28.4

Note: An empty cell indicates that the device did not provide that variable. Absolute bias = mean of the absolute value for bias between devices and PSG (i.e., distance from zero bias).

**Table 5 sensors-22-06317-t005:** Error matrices for classification of sleep state for each device against PSG (%).

		**Apple Watch S6**									
		Wake			Light				Deep		
PSG	Wake	26			51				23		
	N1 or N2	4			44				52		
	N3 or REM	3			26				71		
		**Garmin Forerunner 245 Music**									
		Wake		Light			Deep			REM	
PSG	Wake	27		45			5			23	
	N1 or N2	3		68			8			21	
	N3	1		61			28			10	
	REM	1		42			7			50	
		**Polar Vantage V**									
		Wake		Light			Deep			REM	
PSG	Wake	51		31			5			13	
	N1 or N2	9		60			14			17	
	N3	7		53			33			7	
	REM	5		43			3			49	
		**Oura Ring Generation 2**									
		Wake		Light			Deep			REM	
PSG	Wake	57		30			6			7	
	N1 or N2	9		66			15			10	
	N3	2		32			62			4	
	REM	6		37			5			52	
		**WHOOP 3.0**									
		Wake		Light			Deep			REM	
PSG	Wake	56		28			2			14	
	N1 or N2	12		58			10			20	
	N3	2		32			62			4	
	REM	9		23			2			66	
		**Somfit**									
		Wake	N1			N2		N3			REM
PSG	Wake	57	24			1		6			12
	N1	26	1			53		3			17
	N2	7	1			80		7			5
	N3	5	1			25		68			1
	REM	8	1			26		5			61

Note: Shaded grey cells indicate PSG sleep states were correctly identified by the wearable device.

**Table 6 sensors-22-06317-t006:** Biases, limits of agreement and correlations between ECG and each device for heart rate and heart rate variability (HRV). Biases data are mean ± SD.

Variable	Apple Watch	Garmin	Polar	Oura (Gen.2)	WHOOP (3.0) *	Somfit *
Bias:						
Heart Rate (bpm)	0.5 ± 2.1	5.0 ± 12.8	−1.1 ± 2.2	0.1 ± 4.5	−0.3 ± 1.0	2.2 ± 6.5
HRV (RMSSD, ms)	−9.6 ± 28.1	−22.4 ± 46.9	−8.7 ± 38.0	−10.2 ± 39.4	−4.5 ± 3.9	−20.5 ± 37.2
Absolute Bias:						
Heart Rate (bpm)	1.5 ± 1.5	5.4 ± 12.6	1.5 ± 1.9	1.8 ± 4.1	0.7 ± 0.8	2.6 ± 6.3
HRV (RMSSD, ms)	22.5 ± 19.2	33.1 ± 39.9	18.8 ± 34.0	18.9 ± 35.9	4.7 ± 3.6	24.0 ± 35.0
Limits of Agreement:						
Heart Rate (bpm)	±4.0	±25.0	±4.4	±8.8	±1.9	±12.7
HRV (RMSSD, ms)	±55.2	±92.0	±74.5	±77.2	±7.6	±72.9
Intraclass Correlations:						
Heart Rate (bpm)	0.96	0.41	0.93	0.85	0.99	0.65
HRV (RMSSD, ms)	0.67	0.24	0.65	0.63	0.99	0.69

Note: * Denotes raw data that were provided by the manufacturer. Absolute bias = mean of the absolute value for bias between devices and PSG (i.e., distance from zero bias).

## Data Availability

The datasets generated from the current study are available from the corresponding author on reasonable request.

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
