# Peer review of "A Validation of Six Wearable Devices for Estimating Sleep, Heart Rate and Heart Rate Variability in Healthy Adults"

_sensors, 2022, doi:10.3390/s22166317_

Round 1

Reviewer 1 Report

The authors present the article entitled “A validation of six wearable devices for estimating sleep, heart rate, and heart rate variability in healthy adults”.

This paper examines the validity of six wearable devices (Apple Watch S6, 7 Garmin Forerunner 245 Music, Polar Vantage V, Oura Ring Generation 2, WHOOP 3.0 and Somfit) for assessing sleep, heart rate } variability was also considered. 

The article presents the following concerns:

  • The Abstract section must be restructured. This section should give a pertinent overview of the work in a single paragraph. Please check the guide for authors.

  • The Introduction section must be improved. It presents a lack of background about reported techniques to validate the accuracy of wereables. In this context, it is confused if the article is a state of the art review or a experimental work. Also, the objective of the article is not clear. 

  • Line 76: I suggest to describe what kind of dinner the pacients had (i.e. low carbs). Did all participants were in same dinner conditions?

  • I recommend this reference in line 29, regarding biosensors feedbacking: Speed controller-based fuzzy logic for a biosignal-feedbacked cycloergometer.

  • I recommend this reference in line 58, regarding elderly feedbacked systems in order to introduce the topic of the statistics presented: Elderly Simulation Suit as an Empathic Design Tool.

  • I recommend to add a Conclusion section and place future works in this section.

  • Line 54: What is going to be validated?

  • Lines 230 and 238: Authors mention that some variables were calculated. However, no equation is provided.

  • The final part of the introduction should make a description of the structure of the text.

  • Add a little introduction between points 2 and 2.1. 3 and 3.1, 4 and 4.1

  • Add hyperlinks to tables, figures, and references.

  • In references, the reference 5: The year is missing.

  • Vectorize figures in order to see details.

The following misspelling should be checked:

  1. The Abstract section must be restructured. This section should give a pertinent overview of the work in a single paragraph. Please check the guide for authors.

  2. The Introduction section must be improved. It presents a lack of background about reported techniques to validate the accuracy of wereables. In this context, it is confused if the article is a state of the art review or a experimental work. Also, the objective of the article is not clear. 

  3. Line 54: What is going to be validated?

  4. Lines 230 and 238: Authors mention that some variables were calculated. However, no equation is provided.

  5. line 35: “The most commonly utilised alternative to PSG has previously been actigraphy…” your sentence may be unclear or hard to follow. Consider rephrasing  by “The most commonly utilised PSG alternative has been actigra- 35 PHY [6]”

  6. line 114: “was matched to…” should be rewritten as “matched…”

  7. line 155: “Watch was used to score all time within a sleep period as wake, light sleep or deep sleep.” your sentence may be unclear or hard to follow. Consider rephrasing  by “The watch was used to score all times within a sleep period: wake, light sleep, or deep sleep.”

  8. line 530: “utilise…” should be rewritten as “utilized…”

Author Response

Please see file attached.

Reviewer 2 Report

In this paper, the authors examine the validity of six wearable devices (Apple Watch S6, Garmin Forerunner 245 Music, Polar Vantage V, Oura Ring Generation 2, WHOOP 3.0, and Somfit) for assessing sleep. Validity for heart rate and heart rate variability was also assessed.  Results for the 2-state categorization of sleep indicate that all devices are valid for the field-based assessment of total sleep time. Results for multi-state categorization of sleep indicate that all devices require further improvement for the assessment of specific sleep stages. This article is clear, concise, and suitable for the scope of the journal. Only several small suggestions are supplied:
1. Suggest the authors supply more detail about the Bland-Altman plots for devices and PSG-derived measures of total sleep time.
2. Suggest the authors enhance the introduction part with other technologies for sleep monitoring such as:
Low-cost plastic optical fiber sensor embedded in mattress for sleep performance monitoring, Optical Fiber Technology 64:102541,2021
etc

Author Response

Please see file attached.

Reviewer 3 Report

This paper present study results of six commercial wearable devices (Apple Watch S6, Garmin Forerunner 245 Music, Polar Vantage V, Oura Ring Generation 2, WHOOP 3.0 and Somfit) for sleep, heart rate, and heart rate variability.  The study included 53 healthy adults, who were fitted with all six devices, as well as PSG and ECG as comparison.  Results showed all devices are valid for assessment of total sleep duration, but need improvement for assessment of sleep stages.  

The paper is well organized and section structures are consistent with the proposed objectives.  The selected devices are the most popular ones in the market that makes it attractive to most of audience.  The authors provided sufficient study procedure and participants information.  In results session, the authors descripted performance of each device with detailed data as support.  Figures are clean and easy to understand.  Overall, it is a high quality paper that could be a good contribution to Sensors.

Author Response

Thank you – no changes required based on these comments.

Round 2

Reviewer 1 Report

The manuscript has been greatly improved; I recommend publishing it.